# Three host peculiarities of a cycloalkane-based micelle toward large metal-complex guests

Mamiko Hanafusa[1], Yamato Tsuchida[1], Kyosuke Matsumoto[1], Kei Kondo[1] & Michito Yoshizawa [1✉]

Linear alkanes are essential building blocks for natural and artificial assemblies in water. As compared with typical, linear alkane-based micelles and recent aromatic micelles, we herein develop a cycloalkane-based micelle, consisting of bent amphiphiles with two cyclohexyl frameworks. This uncommon type of micelle, with a spherical core diameter of ~ 2 nm, forms in water in a spontaneous and quantitative manner. The cycloalkane-based, hydrophobic cavity displays peculiar host abilities as follows: (i) highly efficient uptake of sterically demanding Zn(II)-tetraphenylporphyrin and rubrene dyes, (ii) selective uptake of substituted Cu(II)-phthalocyanines and spherical nanocarbons, and (iii) uptake-induced solution-state emission of [Au(I)-dimethylpyrazolate]$_3$ in water. These host functions toward the large metal-complex and other guests studied herein remain unaccomplished by previously reported micelles and supramolecular containers.

[1] Laboratory for Chemistry and Life Science, Institute of Innovative Research, Tokyo Institute of Technology, 4259 Nagatsuta, Midori-ku, Yokohama 226-8503, Japan. ✉email: yoshizawa.m.ac@m.titech.ac.jp

Linear alkanes, which are photo- and electrochemically inactive, act as essential building blocks for natural and artificial amphiphilic molecules. Cell membranes[1] and typical micelles[2,3] are micrometer- and nanometer-sized molecular assemblies, respectively, composed of multiple amphiphiles bearing linear alkyl chains (e.g., Fig. 1a). In contrast, the usability of cyclic alkanes for synthetic as well as biological amphiphiles has been obscure so far, except for steroid-based, oligocyclic amphiphiles (e.g., sodium cholate and CHAPS)[4,5]. The rigidity and directionality of the cyclic frameworks are higher than those of the acyclic ones so that the rational incorporation of cycloalkyl groups into amphiphilic molecules would lead to the development of micelles with unusual host functions, such as enhanced and selective guest uptake[6–16].

For the proof of concept of a cycloalkane-based micelle, we took inspiration from aromatic amphiphile **AA** bearing a bent anthracene dimer with hydrophilic ionic groups (Fig. 1e)[17–23]. The bent amphiphiles spontaneously assemble into aromatic micelle **(AA)**$_n$ in water through π-stacking interactions and the hydrophobic effect (Fig. 1b). Notably, the host capability of **(AA)**$_n$ toward hydrophobic aromatic guests are superior to that of common micelles, for example, sodium dodecyl sulfate-based micelle **(SDS)**$_n$ (Fig. 1d)[17–24]. We thus employed the bent structure and replaced its polyaromatic panels by cycloalkyl groups to make new amphiphile **CHA** (Fig. 1f). Although the intermolecular interactions between cycloalkyl groups are generally weaker than those between polyaromatic rings, our anticipation was that the bent biscycloalkyl framework enables

**CHA** to generate cycloalkane-based micelle **(CHA)**$_n$ (Fig. 1c), with characteristic uptake abilities in water, through the hydrophobic effect and van der Waals interactions. In addition, unlike a majority of metallosupramolecular containers[6–14] and aromatic micelles[17–24], the photoinactive aliphatic shell of **(CHA)**$_n$ was expected not to retard the emission properties of guests even upon incorporation.

We herein report the facile preparation and host functions of cycloalkane-based micelle **(CHA)**$_n$ in water. Bent amphiphiles **CHA** bearing two hydrophobic cyclohexyl and three hydrophilic ammonium groups assemble into a spherical micelle (Fig. 1c, f), with a core diameter of ~2 nm, in a spontaneous and quantitative manner. The flexible cycloalkane-rich cavity demonstrates (i) highly efficient uptake of sterically demanding Zn(II)-tetraphenylporphyrin (**ZnTPP**) and rubrene (**Rub**) dyes, (ii) selective uptake of substituted Cu(II)-phthalocyanines (**CuPc-Cl**) and spherical nanocarbons, and (iii) uptake-induced solution-state emission of [Au(I)-dimethylpyrazolate]$_3$ (**AuPz**) in water. These host peculiarities toward the large metal-complex and other guests studied herein remain unreported with previous micelles as well as supramolecular cages and capsules.

## Results

**Formation of cycloalkane-based micelle (CHA)$_n$.** Cycloalkane-based micelle **(CHA)**$_n$ was spontaneously and quantitatively formed from bent amphiphile **CHA** in water through the hydrophobic effect and van der Waals interactions. The amphiphile was synthesized in four steps starting from the

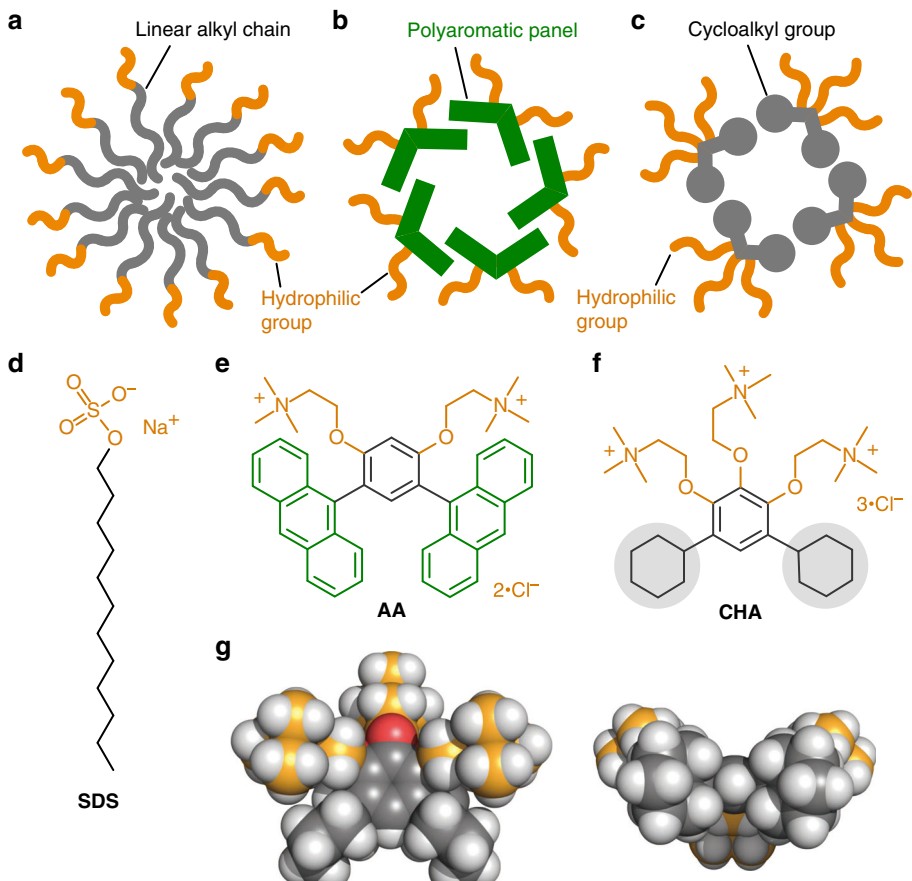

**Fig. 1 Concept and design of a cycloalkane-based micelle.** Schematic representation of **a** a typical micelle, **b** an aromatic micelle, and **c** a micelle with cycloalkyl groups designed in this work. **d** A typical, linear alkane-based amphiphile, **SDS**, **e** bent polyaromatic amphiphile **AA**, and **f** cycloalkane-based, bent amphiphile **CHA**. **g** Side and bottom views of the optimized structure of **CHA** (DFT calculation, B3LYP/6-31G(d) level, white: hydrogen; gray and orange: carbon, red: oxygen, blue: nitrogen).

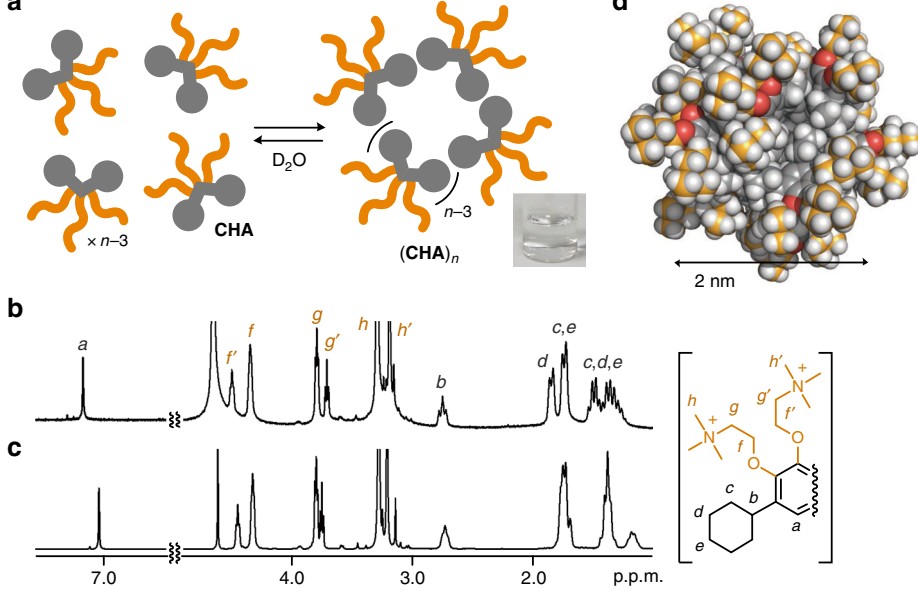

**Fig. 2 Formation of the cycloalkane-based micelle. a** Schematic representation of the quantitative formation of micelle (**CHA**)$_n$ in water and its photograph. Concentration-dependent $^1$H NMR spectra (400 MHz, D$_2$O, room temperature) of (**CHA**)$_n$ at **b** 10 mM and **c** 170 mM based on **CHA**. Labels $a$-$h'$ are the signal assignment of **CHA**. **d** Optimized structure of spherical micelle (**CHA**)$_{12}$ (white: hydrogen; gray and orange: carbon; red: oxygen; blue: nitrogen).

Friedel–Crafts reaction of pyrogallol and chlorocyclohexane, without precious transition-metal catalysts (see "Methods" section). Upon addition of **CHA** (50 μmol) into water (0.3 ml), micelle (**CHA**)$_n$ was instantly generated at room temperature (Fig. 2a). The concentration-dependent proton nuclear magnetic resonance ($^1$H NMR) spectra of **CHA** (from 10 to 170 mM) in D$_2$O showed the upfield shifts of the phenylene and cyclohexyl signals ($H_a$ and $H_{c-e}$; Fig. 2b, c), implying the self-assembly of the hydrophobic moieties. Concentration-dependent dynamic light scattering (DLS) analysis further indicated the quantitative formation of small particles (**CHA**)$_n$ at ≥170 mM and provided an average core diameter of 2.3 nm (see Supplementary Fig. 12). The critical micelle concentration of (**CHA**)$_n$ is ~170 mM, which is much higher than those of (**AA**)$_n$ (~1 mM) and (**SDS**)$_n$ (~10 mM)[17], because of the absence of aromatic π-stacking interactions and/or the presence of the three hydrophilic groups. On the basis of the NMR and DLS analyses, molecular modeling studies suggested a spherical (**CHA**)$_{12}$ structure, having a cycloalkane-based, hydrophobic core with diameters of ~2 nm (Fig. 2d). The present micelle provides faint ultraviolet (UV)–visible absorption bands at ~270 nm (see Supplementary Fig. 13), in contrast to previous aromatic micelle (**AA**)$_n$ with intense absorption bands in the range of <300 and 310–420 nm[17].

**Highly efficient uptake of bulky metal-complex and organic dyes.** Unexpectedly, micelle (**CHA**)$_n$ exhibited enhanced uptake ability toward highly hydrophobic, large dyes **ZnTPP** and **Rub** in water, as compared with previous micelles (**AA**)$_n$ and (**SDS**)$_n$, and consequently generated well water-soluble host–guest complexes. For the uptake of the metal complex, a mixed solid of **CHA** (2.0 μmol) and **ZnTPP** (1.0 μmol) was ground for 6 min using an agate mortar and pestle (Fig. 3a, right). The resultant solid was partially dissolved in H$_2$O (2.0 ml) and the subsequent filtration of the suspension yielded host–guest complex (**CHA**)$_n$•(**ZnTPP**)$_m$ as a clear purple solution. The UV–visible spectrum of the solution displayed intense Soret bands at ~420 nm, derived from the incorporated dyes, without overlap with the host absorption (Fig. 3b). The relative band intensity suggested a 5:1 **CHA**/**ZnTPP** ratio, which was also confirmed by $^1$H NMR

analysis in organic solvent (e.g., DMSO-$d_6$) after the lyophilization of the isolated product (see Supplementary Fig. 16). The DLS chart of the product revealed its average core diameter to be 3.8 nm (Fig. 3d, e), suggesting the formation of a (**CHA**)$_{45}$•(**ZnTPP**)$_9$ complex. In the optimized structure, the sterically demanding **ZnTPP** dyes are fully surrounded by the spherical cycloalkane shell with the multiple hydrophilic pendants (Fig. 3f).

The uptake of **ZnTPP** dyes by (**CHA**)$_n$ was estimated to be 2.5- and 6.7-fold higher than that by aromatic micelle (**AA**)$_n$ and alkane-based micelle (**SDS**)$_n$, respectively, under the same conditions (Fig. 3a, left and 3b). The result most probably stems from the flexible cycloalkane-based cavity capable of interacting with non-planar, bulky surfaces of (**ZnTPP**)$_n$ clusters, through CH–π interactions, to a high degree. In the same way, **Rub** dyes were efficiently uptaken by (**CHA**)$_n$ to give an aqueous red solution with the characteristic bands in the UV–visible spectra (see Supplementary Fig. 18). Host–guest complex (**CHA**)$_n$•(**Rub**)$_m$, featuring an average core diameter of 2.7 nm (see Supplementary Fig. 19), showed apparent absorption bands at ~300 and 420–570 nm (Fig. 3c). Interestingly, the uptake ability of (**CHA**)$_n$ was 2.9 times higher than that of (**AA**)$_n$ toward sterically hindered **Rub** (see Supplementary Fig. 18). It should be noted that there has been no report on micelles as well as supramolecular containers (e.g., large hydrogen-bonding and coordination capsules)[25–27], accommodating such multiple, bulky metal-complex and organic dyes.

**Selective uptake of substituted metal-complexes and spherical nanocarbons.** The polyaromatic cavity of (**AA**)$_n$ has been proven to possess non-selective, wide-ranging host capabilities toward various metallophthalocyanines and various nanocarbons through strong π-stacking interactions[28,29]. In contrast, the present micelle (**CHA**)$_n$ displayed the selective uptake of substituted Cu(II)-phthalocyanines and spherical nanocarbons in water. In a manner similar to the preparation of (**CHA**)$_n$•(**ZnTPP**)$_m$, simple mixing of **CHA** and perchlorinated **CuPc-Cl** (in a 2:1 molar ratio) via manual grinding led to the efficient formation of aqueous host-guest complex (**CHA**)$_n$•(**CuPc-Cl**)$_m$ (Fig. 4a, right). UV–visible spectrum of the resultant green solution showed

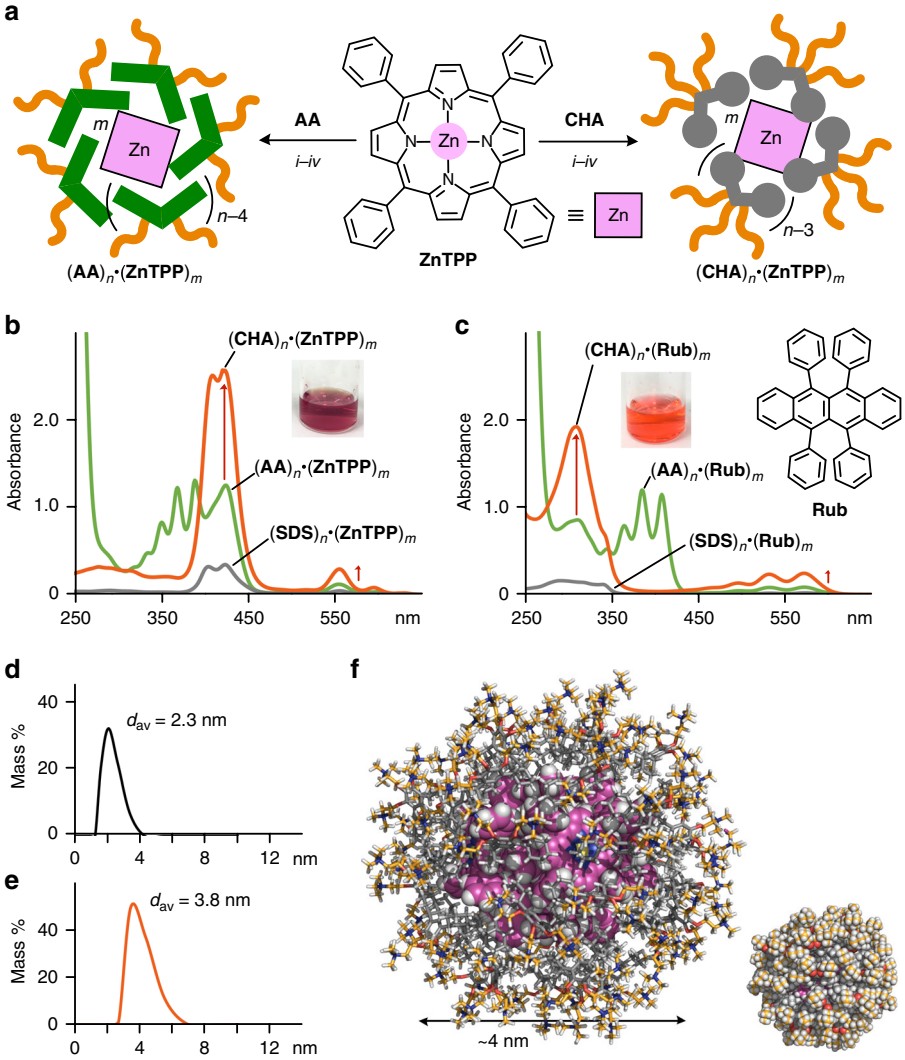

**Fig. 3 Enhanced uptake of bulky metal-complex and organic dyes by the cycloalkane-based micelle. a** Schematic representation of the uptake of **ZnTPP** dyes by (**CHA**)$_n$ or (**AA**)$_n$ through (*i*) grinding (6 min), (*ii*) water addition (2.0 ml), (*iii*) centrifugation (16,000 × *g*, 10 min), and (*iv*) filtration (200 nm pore size). UV–visible spectra and photographs (H$_2$O, room temperature, 1.0 mM based on the amphiphiles) of (**CHA**)$_n$, (**AA**)$_n$, and (**SDS**)$_n$ uptaking **b ZnTPP** and **c Rub** dyes. DLS charts (H$_2$O, room temperature) of **d** (**CHA**)$_n$ and **e** (**CHA**)$_n$•(**ZnTPP**)$_m$ (170 and 1.0 mM based on **CHA**, respectively). **f** Optimized structure of (**CHA**)$_{45}$•(**ZnTPP**)$_9$ (white: hydrogen; gray, orange, and purple: carbon; red: oxygen; blue: nitrogen; yellow: zinc).

broad absorption bands, corresponding to multiply stacked (**CuPc-Cl**)$_m$ within (**CHA**)$_n$, in the range of 290–470 and 530–900 nm (Fig. 4b). The concentration of highly hydrophobic **CuPc-Cl** solubilized in H$_2$O was estimated to be 0.14 mM upon uptake (1.0 mM based on **CHA**). The DLS analysis of (**CHA**)$_n$• (**CuPc-Cl**)$_m$ clarified the formation of only small particles, with their average core diameter being 3.1 nm (see Supplementary Fig. 21).

Remarkably, neither non-substituted **CuPc-H** nor perfluorinated **CuPc-F** was incorporated into (**CHA**)$_n$ under various conditions, for example, with/without grinding and sonication in several amphiphile–substrate ratios (Fig. 4a, left). No absorption bands derived from both **CuPc-H** and **CuPc-F** were detectable in the UV–visible spectra (Fig. 4b). Alkane-based micelles (e.g., (**SDS**)$_n$ and its derivative) show poor uptake abilities and no selectivity toward **CuPc-Cl** under similar conditions[29]. The observed, unusual selectivity is explainable by the self-stacking properties of the planar metal complexes, which are sterically reduced by the slightly larger Cl substituents and distinguished by the cycloalkane-based cavity through multiple CH−π interactions. In the same way, the efficient uptake of bowl-shaped

subphthalocyanine (**SubPc**) dyes was thus accomplished by (**CHA**)$_n$ to afford a violet host–guest complex in water (Fig. 4b and see Supplementary Methods).

The distinction between spherical and tubular nanocarbons was also demonstrated by (**CHA**)$_n$. Like aromatic micelle (**AA**)$_n$, spherical fullerenes **C$_{60}$**, **C$_{70}$**, and **Sc$_3$N@C$_{80}$** were uptaken by (**CHA**)$_n$ in the same way and the resultant, aqueous brown solutions clearly exhibited broadened absorption bands, derived from the incorporated fullerenes (Fig. 4c and see Supplementary Fig. 23). Interestingly, the uptake efficiencies of (**CHA**)$_n$ were enhanced by 1.6 times for **C$_{60}$** and 1.5 times for **C$_{70}$**, as compared with those of (**AA**)$_n$ under the same conditions. The detailed NMR integral and DLS analyses of the former product suggested the selective formation of a (**CHA**)$_{10}$•(**C$_{60}$**)$_4$ complex, being different from 1:1 host–guest complex (**AA**)$_5$•**C$_{60}$** (see Supplementary Fig. 25)[28]. Whereas a huge number of fullerene-based 1:1 host–guest complexes has been reported so far, the facile preparation of host•(fullerene)$_m$ complexes (*m* ≥ 2) remains challenging[30–32]. It is worth to note that, unlike (**AA**)$_n$, single-walled carbon nanotubes (**CNT**; 0.7–0.9 nm in diameter and ~0.7 μm in length) were not incorporated by (**CHA**)$_n$ so that no

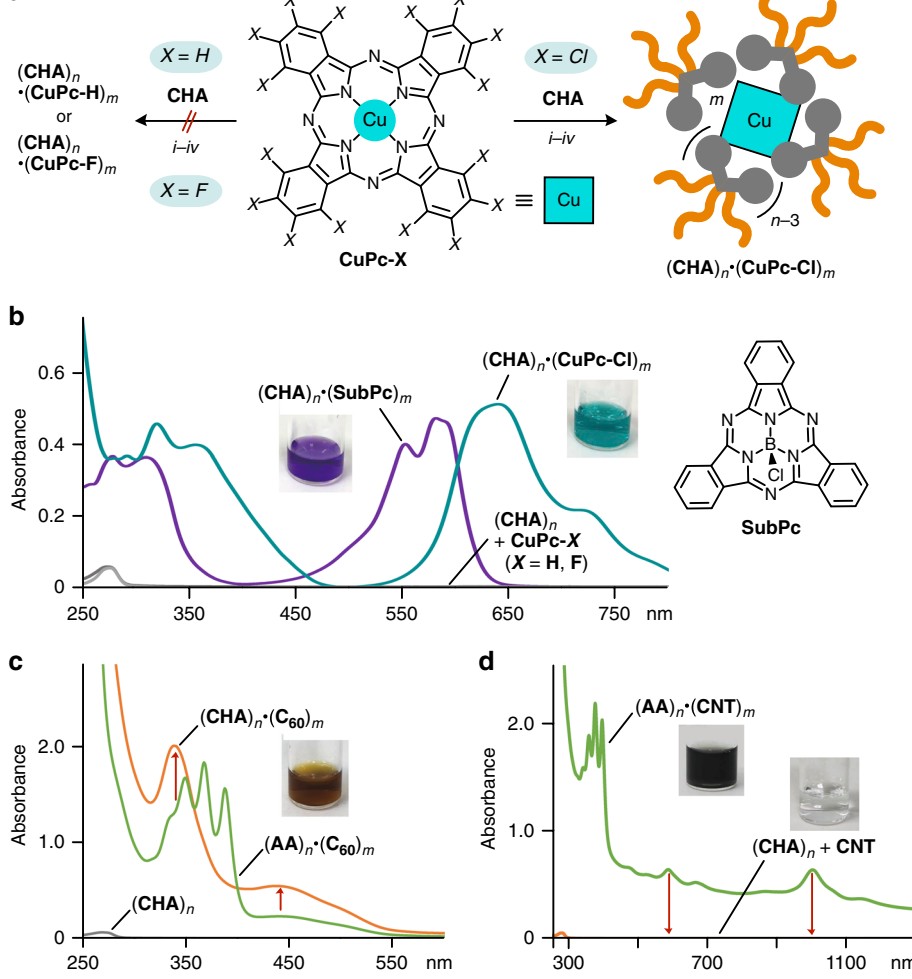

**Fig. 4 Selective uptake of substituted metal-complexes and spherical nanocarbons by the cycloalkane-based micelle. a** Schematic representation of the uptake of **CuPc-X** (X = Cl) by (**CHA**)$_n$ through (*i*) grinding, (*ii*) water addition, (*iii*) centrifugation, and (*iv*) filtration. UV–visible spectra and photographs (H$_2$O, room temperature, 1.0 mM based on the amphiphiles) of **b** (**CHA**)$_n$•(**CuPc-Cl**)$_m$, (**CHA**)$_n$•(**SubPc**)$_m$, and products after treatment of (**CHA**)$_n$ with **CuPc-X** (X = H, F), **c** (**CHA**)$_n$•(**C$_{60}$**)$_m$ and (**AA**)$_n$•(**C$_{60}$**)$_m$, and **d** (**AA**)$_n$•(**CNT**)$_m$ and products after treatment of (**CHA**)$_n$ with **CNT**.

aqueous solution in black was obtained even under the optimized procedure for the preparation of (**AA**)$_n$•(**CNT**)$_m$ (i.e., 6 min grinding and 15 min sonication; Fig. 4d). The difference in self-aggregating ability of the nanocarbones most probably causes its shape-selective uptake by (**CHA**)$_n$ in water.

**Uptake-induced solution-state emission of metal-complexes.**
Finally, the facile synthesis of water-soluble host–guest complexes with strong red emission was succeeded using **CHA** and tri-nuclear Au(I) complexes. **AuPz** is known for being highly emissive only in the solid state, due to the indispensable inter-molecular Au(I)•••Au(I) interactions (Fig. 5a, left)[33–35]. To the best of our knowledge, the solution-state emission of trinuclear Au(I) complexes is rare and there has been no report on emissive host–guest complexes including multiple **AuPz** compounds and its derivatives so far[36–40]. When the standard uptake protocol was applied to a mixture of **CHA** and **AuPz** (in a 2:1 molar ratio), colorless host–guest complex (**CHA**)$_n$•(**AuPz**)$_m$ was obtained as a clear aqueous solution (Fig. 5a, right). The UV–visible spectrum showed broad absorption bands, for (**AuPz**)$_m$ incorporated into (**CHA**)$_n$, in the short wavelength region (<310 nm; Fig. 5b). DLS analysis of the product ($d$ = 3.1 nm) also indicated the successful uptake of (**AuPz**)$_m$ ($m$ = ~12) by (**CHA**)$_n$ in water (Fig. 5c, d).

On the other hand, no and low uptake of (**AuPz**)$_m$ were observed with previous micelles (**AA**)$_n$ and (**SDS**)$_n$, respectively, under the same conditions (see Supplementary Fig. 28). The UV–visible analysis revealed that the **AuPz** uptake by (**CHA**)$_n$ is 5.3-fold enhanced over that by (**SDS**)$_n$.

The aqueous solution of (**CHA**)$_n$•(**AuPz**)$_m$ in hand emitted strong red phosphorescence upon irradiation at 290 nm at room temperature, whereas no emission was observed from free **AuPz** in CHCl$_3$ (Fig. 5e). The emission spectrum showed intense broad bands at $\lambda_{max}$ = 711 nm, assignable to typical aurophilic interac-tions[33–35]. The emission bands are slightly hypsochromically shifted ($\Delta\lambda$ = ~15 nm) as compared with that of **AuPz** in the solid state. Although the emission quantum yield of solid **AuPz** ($\Phi$ = 84%) is significantly high, ~40% of the emissivity was retained even in aqueous solution through efficient uptake of (**AuPz**)$_m$ by (**CHA**)$_n$ ($\Phi$ = 32%). The emission band and quantum yield of host–guest complex (**CHA**)$_n$•(**AuPz**)$_m$ are insensitive to air in water (see Supplementary Fig. 30). The CIE chromaticity diagram of **AuPz** within and without (**CHA**)$_n$ was used to quantify the total emission color ((x, y = 0.57, 0.33) and (x, y = 0.61, 0.37), respectively; Fig. 5f). The long emission lifetime of (**CHA**)$_n$•(**AuPz**)$_m$ ($\tau$ = 14.4 µs) supported its phos-phorescence derived from intermolecular Au(I)•••Au(I) interac-tions (Fig. 5g). The emission intensity of the product is 3.8 times

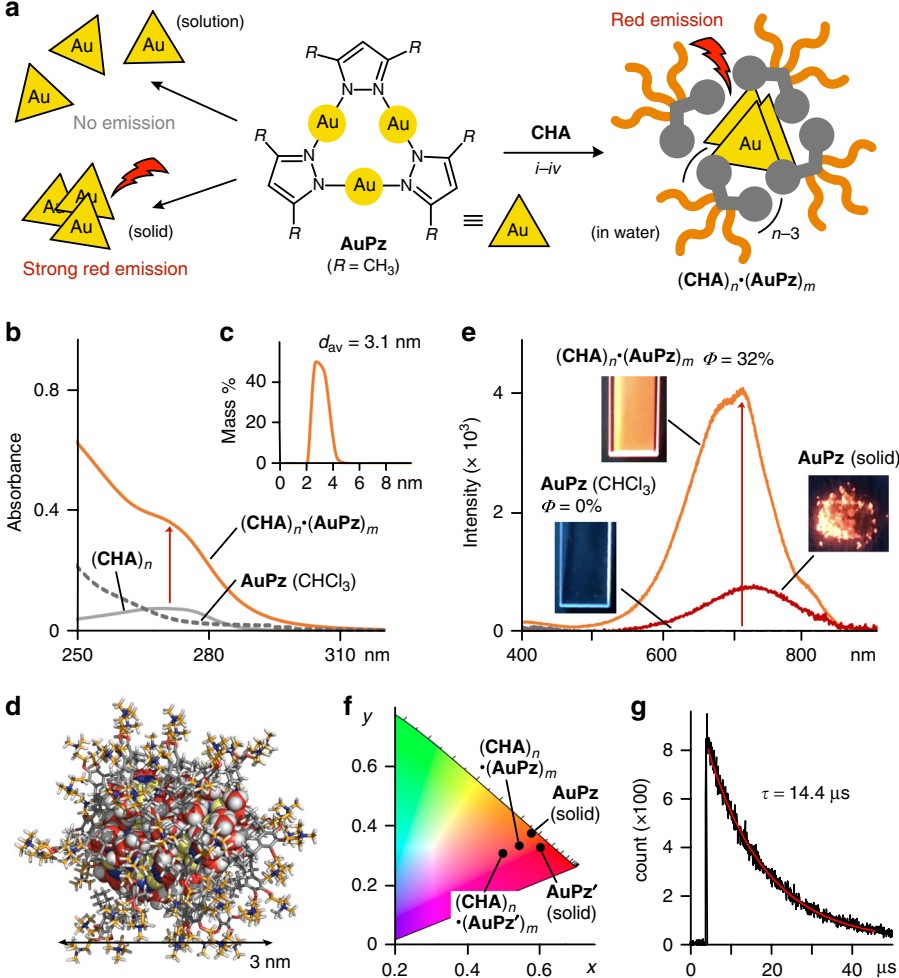

**Fig. 5 Uptake-induced solution-state emission of metal-complexes by the cycloalkane-based micelle. a** Schematic representation of the emission properties of **AuPz** in solution and the solid state, and upon uptake by **(CHA)**$_n$ in water through (*i*) grinding, (*ii*) water addition, (*iii*) centrifugation, and (*iv*) filtration. **b** UV–visible spectra (H$_2$O, room temperature, 1.0 mM based on the amphiphiles) of **(CHA)**$_n$•**(AuPz)**$_m$, **(CHA)**$_n$, and **AuPz** in CHCl$_3$ (0.1 mM). **c** DLS chart of **(CHA)**$_n$•**(AuPz)**$_m$ and **d** optimized structure of **(CHA)**$_{22}$•**(AuPz)**$_{12}$ (white: hydrogen; gray, orange, and scarlet: carbon, red: oxygen, blue: nitrogen, yellow: gold). **e** Emission spectra (room temperature, $\lambda_{ex} = 290$ nm) and photographs ($\lambda_{ex} = 254$ nm) of **(CHA)**$_n$•**(AuPz)**$_m$ in H$_2$O, and **AuPz** in CHCl$_3$ (0.1 mM) and the solid state. **f** CIE coordinate diagram (H$_2$O, room temperature) of **(CHA)**$_n$•**(AuPz)**$_m$, **(CHA)**$_n$•**(AuPz′)**$_m$, **AuPz**, and **AuPz′**. **g** Emission decay profile (H$_2$O, room temperature, $\lambda_{ex} = 280$ nm, $\lambda_{det} = 700$ nm) of **(CHA)**$_n$•**(AuPz)**$_m$.

higher than that of **(SDS)**$_n$•**(AuPz)**$_m$ (see Supplementary Fig. 30). The same procedure also gave rise to red-emissive host–guest complex **(CHA)**$_n$•**(AuPz′)**$_m$ ($\lambda_{max} = 670$ nm, $\Phi = 15\%$) from **CHA** and non-substituted **AuPz′** (R = H; see Supplementary Fig. 32)[34], whereas the trinuclear Au(I)-complex ($\Phi = 49\%$ in the solid state) is insoluble in common organic solvents. Therefore, strong red emission of otherwise solution-state non-emissive **AuPz** and **AuPz′** was demonstrated upon uptake by **(CHA)**$_n$ in aqueous solution.

cycloalkane-based micelle. The obtained host-guest complexes provide multiple, large metal-complexes in the cavity, which is also uncommon for previously reported micelles and supramolecular containers. On the basis of the present and our previous studies[24], we herein emphasize that bent frameworks composed of not only (poly)aromatic panels but also cycloalkane groups are of vital importance for the design of micellar systems with intriguing host functions, which will provide further potentials in host-guest chemistry.

## Discussion

To explore unusual host functions of micellar systems, we have synthesized a cycloalkane-based bent amphiphile, as a relative of a bent polyaromatic amphiphile. In water, the present amphiphile generated a new micelle, with a flexible cavity surrounded by a cycloalkane-rich spherical shell, in a quantitative manner. Investigation of the host ability toward metal-complex guests successfully elucidated the following three peculiar features: enhanced uptake of bulky Zn(II)-complexes, selective uptake of substituted planar Cu(II)-complexes, and uptake-induced solution-state emission of trinuclear Au(I)-complexes by the

## Methods

**General**. NMR: Bruker AVANCE-400 and 500 (400 and 500 MHz); Matrix-assisted laser desorption ionization-time of flight mass spectrometry (MALDI-TOF MS): Bruker UltrafleXtreme; electrospray ionization-TOF-MS: Bruker microTOF II; Fourier transform infrared spectroscopy (FT-IR): SHIMADZU IRSpirit-T; DLS: Wyatt Technology DynaPro NanoStar; UV–visible: JASCO V-670DS; emission: Hitachi F7000; Absolute emission quantum yield: Hamamatsu Quantaurus-QY C11347-01; emission lifetime: Hamamatsu Quantaurus-Tau C11367. Density functional theory (DFT) calculation: Wavefunction, Inc., Spartan'10; molecular mechanics calculation (geometry optimization): Dassault Systèmes Co., Materials Studio, Forcite module (version 5.5.3). Solvents, reagents, and guests (e.g., **ZnTPP**, **Rub**, **CuPc-Cl**, **SubPc**, and **C$_{60}$**): TCI Co., Ltd., FUJIFILM Wako Chemical Co., Kanto Chemical Co., Inc., Sigma-Aldrich Co., and Cambridge Isotope Laboratories, Inc. Anthracene-based amphiphile **AA** and chloro(tetrahydrothiophene)Au(I)

were synthesized according to previously reported procedures (see Supplementary Methods)[17,28,34].

**Synthesis of 1,5-dicyclohexyl-2,3,4-trihydroxybenzene.** Pyrogallol (3.556 g, 28.22 mmol), chlorocyclohexane (10.03 g, 84.58 mmol), and iron(III) chloride hexahydrate (132 mg, 0.488 mmol) were added to a 50-ml glass flask[41]. The resultant mixture was stirred at 120 °C for 12 h. Water (50 ml) was added to the residue and then the mixture was extracted with diethyl ether ($3 \times 50$ ml). The combined organic layers were dried over $MgSO_4$, filtered, and concentrated under reduced pressure. The obtained solid was purified by silica-gel column chromatography ($CH_2Cl_2$) to afford 1,5-dicyclohexyl-2,3,4-trihydroxybenzene (3.492 g, 12.02 mmol, 43% yield) as a white solid. See Supplementary Figs. 1–4.

$^1$H NMR (400 MHz, acetone-$d_6$, room temperature): δ 1.28–1.41 (m, 10H), 1.72–1.79 (m, 10H), 2.84 (t, 2H, $J = 8.8$ Hz), 6.51 (s, 1H), 6.91 (br, 3H). $^{13}$C NMR (100 MHz, acetone-$d_6$, room temperature): δ 27.2 ($CH_2$), 27.9 ($CH_2$), 34.3 ($CH_2$), 38.1 (CH), 115.3 (CH), 126.6 ($C_q$), 133.4 ($C_q$), 141.7 ($C_q$). FT-IR (KBr, cm$^{-1}$): 2924, 2850, 1500, 1450, 1288, 1230, 1092, 976, 575. HR MS (ESI): calcd. For $C_{18}H_{26}O_3Na$ 313.1774 $[M + Na]^+$, found 313.1771.

**Synthesis of CHA.** 1,5-Dicyclohexyl-2,3,4-trihydroxybenzene (3.308 g, 11.39 mmol), NaOH (4.724 g, 118.1 mmol), 2-chloro-$N,N$-dimethylethanamine hydrochloride (7.412 g, 51.46 mmol), and toluene (50 ml) were added to a two-necked 100-ml glass flask filled with $N_2$. The resultant mixture was stirred at 130 °C for 12 h. Water (50 ml) was added to the resulting solution and then the two layers were separated. The organic layer was washed with water ($2 \times 50$ ml), dried over $Na_2SO_4$, filtered, and concentrated under reduced pressure to afford a brown liquid[17]. This liquid was used directly for the next reaction without further purification. The liquid and $CH_3CN$ (50 ml) were added to a 50 ml glass flask. $CH_3I$ (3.5 ml, 56 mmol) was added dropwise to this flask and the resultant mixture was stirred at room temperature overnight. The precipitated crude product was separated and washed with acetonitrile ($2 \times 20$ ml) to afford a white solid (4.884 g, 5.254 mmol). The solid and AgCl (3.001 g, 20.94 mmol) were stirred in $H_2O$ (20 ml) at 80 °C for 12 h. After the addition of $CH_3OH$ (20 ml), the resultant solution was filtered and concentrated under vacuum to afford **CHA** (2.643 g, 2.034 mmol; 35% yield) as a white solid[17]. See Supplementary Figs. 5–9.

$^1$H NMR (400 MHz, $D_2O$, 2 mM, room temperature): δ 1.21–1.49 (m, 10H), 1.70–1.82 (m, 10H), 2.84 (t, 2H, $J = 8.8$ Hz), 3.21 (s, 9H), 3.31 (s, 18H), 3.75 (br, 2H), 3.82 (br, 4H), 4.38 (br, 4H), 4.52 (br, 2H), 7.08 (s, 1H). $^{13}$C NMR (100 MHz, $D_2O$, 2 mM, room temperature): δ 25.7 ($CH_2$), 26.7 ($CH_2$), 33.6 ($CH_2$), 37.7 ($CH_2$), 54.0 ($CH_3$), 54.3 ($CH_3$), 65.0 ($CH_2$), 65.9 ($CH_2$), 66.8 ($CH_2$), 67.7 ($CH_2$), 120.7 (CH), 138.7 ($C_q$), 143.2 ($C_q$), 146.6 ($C_q$). FT-IR (KBr, cm$^{-1}$): 3016, 2927, 2850, 1631, 1477, 1442, 1315, 1049, 953, 532. ESI-TOF MS ($CH_3OH$): $m/z$ 618.5 $[M-Cl^-]^+$, 291.8 $[M-2 \cdot Cl^-]^{2+}$, 182.8 $[M-3 \cdot Cl^-]^{3+}$.

**Formation of micelle $(CHA)_n$.** Amphiphile **CHA** (85 μmol) was added to water (0.5 ml) and the solution was stirred at room temperature for 1 min to give micelle $(CHA)_n$. The resultant clear solution was analyzed by $^1$H NMR, UV–vis, fluorescence, and DLS instruments. The concentration-dependent $^1$H NMR and DLS analyses of $(CHA)_n$ (10, 100, 170, and/or 300 mM based on **CHA**) were also examined in water at room temperature. The optimized structure of micelle $(CHA)_{12}$ was obtained by molecular mechanics calculation (forcite module, Materials Studio). See Supplementary Figs. 10–13.

$^1$H NMR (400 MHz, $D_2O$, room temperature, 170 mM based on **CHA**): δ 1.16 (br, 2H), 1.32–1.44 (br, 8H), 1.70–1.78 (m, 10H), 2.76 (br, 2H), 3.24 (s, 9H), 3.31 (s, 18H), 3.80 (t, 2H, $J = 5.4$ Hz), 3.85 (t, 4H, $J = 5.4$ Hz), 4.38 (br, 4H), 4.51 (t, 2H, $J = 5.4$ Hz), 7.00 (s, 1H). $^{13}$C NMR (125 MHz, $D_2O$, room temperature, 170 mM based on **CHA**): δ 25.9 ($CH_2$), 26.7 ($CH_2$), 33.9 ($CH_2$), 37.6 ($CH_2$), 54.0 ($CH_3$), 54.3 ($CH_3$), 65.1 ($CH_2$), 65.9 ($CH_2$), 67.0 ($CH_2$), 67.8 ($CH_2$), 120.3 (CH), 138.6 ($C_q$), 143.5 ($C_q$), 146.7 ($C_q$).

**Formation of $(CHA)_n \bullet (ZnTPP)_m$.** A mixture of **CHA** (1.3 mg, 2.0 μmol) and **ZnTPP** (0.7 mg, 1.0 μmol) was ground for 6 min using an agate mortar and pestle. After the addition of $H_2O$ (2.0 ml) to the mixture, the suspended solution was centrifuged ($16,000 \times g$, 10 min) and then filtered by a membrane filter (pore size: 200 nm) to give a clear purple solution of $(CHA)_n \bullet (ZnTPP)_m$. The structure of $(CHA)_n \bullet (ZnTPP)_m$ was confirmed by UV–visible and DLS analyses. The averaged host–guest stoichiometry for the product (i.e., $n = 45$ and $m = 9$) and the concentration of encapsulated **ZnTPP** (0.2 mM) were estimated by the $^1$H NMR analysis of the product in DMSO-$d_6$. The same procedure using **AA** (1.4 mg, 2.0 μmol) and **ZnTPP** (0.7 mg, 1.0 μmol) or **SDS** (0.6 mg, 2.0 μmol) and **ZnTPP** (0.7 mg, 1.0 μmol) afforded a clear purple solution of $(AA)_n \bullet (ZnTPP)_m$ or $(SDS)_n \bullet (ZnTPP)_m$. See Supplementary Figs. 14–17.

The pairwise uptake experiment of **ZnTPP** and **Rub** by micelle $(CHA)_n$ led to the formation of a mixture of host–guest complexes (see Supplementary Fig. 34).

**Formation of $(CHA)_n \bullet (CuPc-Cl)_m$.** A mixture of **CHA** (1.3 mg, 2.0 μmol) and perchlorinated Cu(II)-phthalocyanine (**CuPc-Cl**; 1.1 mg, 1.0 μmol) was ground for 6 min using an agate mortar and pestle[29]. After the addition of $H_2O$ (2.0 ml) to the

mixture, the suspended solution was centrifuged ($16,000 \times g$, 10 min) and then filtrated by a membrane filter (pore size: 200 nm) to give a clear green solution of $(CHA)_n \bullet (CuPc-Cl)_m$. The structure of $(CHA)_n \bullet (CuPc-Cl)_m$ was confirmed by UV–visible and DLS analyses. The concentration of encapsulated **CuPc-Cl** was estimated to be 0.14 mM by the UV–visible analysis. In the same way, a 2:1 mixture of **CHA** and non-substituted Cu(II)-phthalocyanine (**CuPc-H**) or perfluorinated Cu(II)-phthalocyanine (**CuPc-F**) was ground for 6 min. After the addition of $H_2O$ (2.0 ml) to the mixture, the suspended solutions were centrifuged ($16,000 \times g$, 10 min) and then filtered by a membrane filter (pore size: 200 nm) to give colorless solutions of guest-free micelle $(CHA)_n$, as confirmed by UV–visible analysis. See Supplementary Figs. 20–22.

**Formation of $(CHA)_n \bullet (C60)_m$.** A mixture of **CHA** (1.3 mg, 2.0 μmol) and $C_{60}$ (1.5 mg, 2.0 μmol) was ground for 6 min using an agate mortar and pestle[28]. After the addition of $H_2O$ (2.0 ml) to the mixture, the suspended solution was centrifuged ($16,000 \times g$, 10 min) and then filtered by a membrane filter (pore size: 200 nm) to give a clear brown solution of $(CHA)_n \bullet (C_{60})_m$. The structure of host–guest complex $(CHA)_n \bullet (C_{60})_m$ was confirmed by UV–visible and DLS analyses. The average host–guest stoichiometry for the product (i.e., $(CHA)_{10} \bullet (C_{60})_4$) and the concentration of encapsulated $C_{60}$ (0.4 mM) were estimated by UV–visible analysis with a calibration curve method in toluene after the lyophilization of the isolated product. In the same way, host–guest complexes $(CHA)_n \bullet (C_{70})_m$ and $(CHA)_n \bullet (Sc_3N@C_{80})_m$ were prepared using **CHA** (1.3 mg, 2.0 μmol) and $C_{70}$ (1.7 mg, 2.0 μmol) or **CHA** (0.7 mg, 1.0 μmol) and $Sc_3N@C_{80}$ (0.4 mg, 0.3 μmol). Clear brown solutions of $(AA)_n \bullet (C_{60})_m$, $(AA)_n \bullet (C_{70})_m$, and $(AA)_n \bullet (Sc_3N@C_{80})_m$ were obtained by the same procedure using **AA** and the corresponding fullerenes. A mixture of **CHA** (1.3 mg, 2.0 μmol) and single-walled CNTs (0.6 mg; 0.7–0.9 nm in diameter) was ground for 6 min using an agate mortar and pestle[28]. After the addition of $H_2O$ (2.0 ml), the suspended mixture was sonicated (40 kHz, 150 W) for 15 min at room temperature. The centrifugation ($16,000 \times g$, 10 min) of the resultant solution afforded a colorless solution of guest-free micelle $(CHA)_n$, as confirmed by UV–visible–NIR analysis. In contrast, a clear black solution of $(AA)_n \bullet (CNT)_m$ was obtained by the same procedure using **AA** (1.5 mg, 2.1 μmol) and **CNT** (0.6 mg). See Supplementary Figs. 23–26.

The pairwise uptake experiment of $C_{60}$ and **CNT** by micelle $(CHA)_n$ led to the formation of a mixture of host–guest complexes (see Supplementary Fig. 34).

**Formation of $(CHA)_n \bullet (AuPz)_m$.** A mixture of **CHA** (1.3 mg, 2.0 μmol) and **AuPz** (0.9 mg, 1.0 μmol)[34] was ground for 6 min using an agate mortar and pestle. After the addition of $H_2O$ (2.0 ml), the suspended solution was centrifuged ($16,000 \times g$, 10 min) and then filtered by a membrane filter (pore size: 200 nm) to give a colorless solution of $(CHA)_n \bullet (AuPz)_m$. The structure of host–guest complex $(CHA)_n \bullet (AuPz)_m$ was confirmed by UV–visible, emission, and DLS analyses. Host–guest complex $(SDS)_n \bullet (AuPz)_m$ was obtained in the same way. In contrast, the same procedure using **AA** and **AuPz** gave only guest-free micelle $(AA)_n$. Host–guest complex $(CHA)_n \bullet (AuPz')_m$ was obtained in the same way from a mixture of **CHA** and non-substituted **AuPz'**. See Supplementary Figs. 27–33.

## Data availability

The authors declare that the data supporting the findings of this study are available within the Supplementary Information file and from the corresponding author upon reasonable request.

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

## Acknowledgements

We acknowledge the financial support from JSPS KAKENHI (Grant No. JP17H05359/JP18H01990/JP19H04566) and "Support for Tokyo Tech Advanced Researchers (STAR)." We thank Dr. Takamasa Tsukamoto and Prof. Kimihisa Yamamoto (Tokyo Institute of Technology) for supporting emission lifetime analysis.

## Author contributions

M.H., Y.T., K.M., K.K., and M.Y. designed the work, carried out research, analyzed data, and wrote the paper. M.Y. is the principal investigator. All authors discussed the results and commented on the manuscript.

## Competing interests

The authors declare no competing interests.

## Additional information

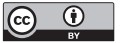

