## [Peer Review File · Nature Communications]

REVIEWER COMMENTS

Reviewer #1 (Remarks to the Author):

Yoshizawa and co-workers report the spontaneous assembly of micelles based on the cycloalkane amphiphile CHA and demonstrate the unusual guest uptake properties of the (CHA)_n micelles compared to their previously reported micelles based on aromatic amphiphiles (AA). Guest uptake could be more easily monitored by UV/vis spectroscopy since the (CHA)_n micelles have very weak absorption bands compared to the intense bands of the aromatic micelles, which can overlap with the guests. While the aromatic micelles showed non-selective guest uptake, the (CHA)_n micelles included perchlorinated Cu(II) phthalocyanines but not their perfluorinated or non-substituted analogues, attributed to the steric bulk of the chlorine substituents reducing self-stacking. Furthermore, the (CHA)_n micelles showed shape selectivity since only uptake of spherical nanocarbons (C₆₀, C₇₀ and Sc₃N@C₈₀) but not single-walled carbon nanotubes was observed. In addition to the improved selectivity, the (CHA)_n micelles had a 1.5-3x better uptake of the guests compared to the aromatic micelles. Finally, the emission of the trinuclear AuPz complex was demonstrated in solution when it is normally only emissive in the solid-state as a result of intermolecular Au-Au interactions.

This is an interesting and well-performed study that would be of interest to the wide readership of Nature Communications. The authors demonstrate that (CHA)_n micelles are an interesting new class of micelles with improved uptake and selectivity over related classes of micelles like aromatic micelles. I believe the results of this manuscript will stimulate further investigation on this new class of micelles and I would recommend publication of the manuscript with the following minor corrections.

Figure 3A could be improved to make it clearer that the uptake of ZnTPP and Rub by (CHA)_n was being investigated by showing a general host-guest complex. In its current form, the meaning of the thick black line separating Rub and the host-guest complex with ZnTPP is not clear.

In the SI, the ¹H NMR data for multiplets should be consistently reported as ranges.

The authors present individual experiments demonstrating that the (CHA)_n micelles only include certain guests, e.g. spherical but not tubular nanocarbons. Have the authors performed analogous experiments with mixtures of the guests? While I believe the manuscript can be accepted in its current form, inclusion of these experiments in the manuscript (if they have already been performed) would nicely demonstrate the selectivity and potential application of the micelles in separations.

Reviewer #2 (Remarks to the Author):

This is a very interesting paper revealing novel types of micelles formed by small amphiphilic molecules with cycloalkane core segments. The micelles have unique complexation capabilities toward metal-containing molecules, allowing to preserve and further transform their photo-emission properties. Given the unusual properties that these micelles have, it might be useful to reveal the structures of these self-assemblies through simulations or analytical tools, to better understand the origin of their properties. Do the complexed micelles behave as plasmonic nanoparticles? Could the authors potentially address these issues? If these changes are made, I believe that the paper can be published in Nature Communications.

Reviewer #3 (Remarks to the Author):

The authors describe synthesis of a new amphiphile (CHA) decorated with two cyclohexyl groups, and its solvation behavior for a variety of organic and coordination compounds in water. The uptake properties of CHA are different from those of AA, in particular, CHA can solvate guest aggregates and does not disturb photo-excitation of the aggregates. This system is well-designed on the basis of authors' previous amphiphile (AA) having two aromatic groups and will extend their molecular concept. However, I would hesitate an acceptance of this paper at a present form, because this paper consists of a simple comparison about uptake properties with author's aromatic micelle. In order to improve the novelty of this paper, I would suggest that the authors should add new results and mention more advantages by usage of this new amphiphile.

The CHA amphiphile can uptake guest aggregates rather than a single molecule in water, and thereby they are expected to show remarkable properties far different from their solid and solution states. I would recommend the authors to show unusual behavior originated from the guest aggregates in the CHA micelle. For example, the authors speculated that four C60 molecules were trapped together in the CHA amphiphile. Does the C60 aggregate show unusual redox behavior? The CHA amphiphile can create a large hydrophobic space in water, where can accommodate several guest molecules. Did the authors try co-encapsulation of two different guests? They will establish new triplet-triplet annihilation and donor-acceptor systems.

For the comments of Reviewer #1:

Yoshizawa and co-workers report the spontaneous assembly of micelles based on the cycloalkane amphiphile CHA and demonstrate the unusual guest uptake properties... This is an interesting and well-performed study that would be of interest to the wide readership of Nature Communications. The authors demonstrate that (CHA)_n micelles are an interesting new class of micelles with improved uptake and selectivity over related classes of micelles like aromatic micelles. I believe the results of this manuscript will stimulate further investigation on this new class of micelles and I would recommend publication of the manuscript with the following minor corrections.

We appreciate having a very positive evaluation of our present results from Reviewer #1 after careful reading of all over our manuscript.

1) Figure 3A could be improved to make it clearer that the uptake of ZnTPP and Rub by (CHA)_n was being investigated by showing a general host-guest complex. In its current form, the meaning of the thick black line separating Rub and the host-guest complex with ZnTPP is not clear.

According to the reviewer's kind suggestion, we modified Fig. 3a as follows: the chemical structure of **Rub** was moved to Fig. 3c and then the formation of (AA)_n•(ZnTPP)_m was added to Fig. 3a, left.

2) In the SI, the ¹H NMR data for multiplets should be consistently reported as ranges.

There are two types of multiplets in ¹H NMR peaks: one is multiply split peaks derived from environmentally "equivalent", single protons and other is overlapped, several peaks derived from environmentally "unequivalent", several protons. We have reported the chemical shifts of center peaks for the first type and the ranges of chemical shifts for the second type in our SI.

3) The authors present individual experiments demonstrating that the (CHA)_n micelles only include certain guests, e.g. spherical but not tubular nanocarbons. Have the authors performed analogous experiments with mixtures of the guests? While I believe the manuscript can be accepted in its current form, inclusion of these experiments in the manuscript (if they have already been performed) would nicely demonstrate the selectivity and potential application of the micelles in separations.

This reviewer has fully understood the present system. Yes, we had already examined the suggested uptake experiment, using a mixture of fullerene C₆₀ and single-walled carbon nanotubes (CNT). Thus, we added the following sentence to the Methods section (page 12): "*The pairwise uptake experiment of C₆₀ and CNT by micelle (CHA)_n led to the formation of a mixture of host-guest complexes (See Supplementary Methods and Fig. 34).*" However, we need to improve the uptake efficiency and characterize the detailed host-guest structures, before discussion of the potential application.

Furthermore, we additionally examined the uptake experiments of micelle (CHA)_n from mixtures of ZnTPP/Rub, ZnTPP/CuPc-Cl, ZnTPP/C₆₀, and C₆₀/SubPc, respectively. In a manner similar to the C₆₀/CNT study, the UV-visible analysis of the products indicated the formation of a complex mixture of host-guest complexes (Supplementary Fig. 34).

For the comments of Reviewer #2:

This is a very interesting paper revealing novel types of micelles... Given the unusual properties that these micelles have, it might be useful to reveal the structures of these self-assemblies through simulations or analytical tools, to better understand the origin of their properties. Do the complexed micelles behave as plasmonic nanoparticles? Could the authors potentially address these issues? If these changes are made, I believe that the paper can be published in Nature Communications.

We again appreciate having a very positive evaluation of our work from Reviewer #2.

Usually, nanoparticles show plasmonic properties in the range of $\sim 10^2$ nm dimension. On the other hand, the size of the present micelle and its host-guest complexes is approximately 2-4 nm so that we cannot observe its plasmon. Encapsulation of pristine nanoparticles ($\sim 10^2$ nm in diameter) by our micelles is one of our next targets for the facile preparation of aqueous plasmonic nanoparticles.

For the comments of Reviewer #3:

The authors describe synthesis of a new amphiphile (CHA) decorated with two cyclohexyl groups, and its solvation behavior for a variety of organic and coordination compounds in water. The uptake properties of CHA are different from those of AA, in particular, CHA can solvate guest aggregates and does not disturb photo-excitation of the aggregates. This system is well-designed on the basis of authors' previous amphiphile (AA) having two aromatic groups and will extend their molecular concept.

We appreciate having a positive evaluation of our well-designed system from Reviewer #3.

However, I would hesitate an acceptance of this paper at a present form, because this paper consists of a simple comparison about uptake properties with author's aromatic micelle.

In this work, we compared the uptake abilities of new cycloalkyl micelle (**CHA**)_n with those of our previous micelle (**AA**)_n as well as typical alkyl micelle (**SDS**)_n, and then revealed its excellent uptake ability. To avoid this misleading, we added the chemical structure of **SDS** to Fig. 1d and "as compared with previous micelles (**AA**)_n and (**SDS**)_n" to the revised text (page 4).

In order to improve the novelty of this paper, I would suggest that the authors should add new results and mention more advantages by usage of this new amphiphile.

The CHA amphiphile can uptake guest aggregates rather than a single molecule in water, and thereby they are expected to show remarkable properties far different from their solid and solution states. I would recommend the authors to show unusual behavior originated from the guest aggregates in the CHA micelle.

In response to these comments from Reviewer #3, here we would like to emphasize the novelty of our present results and explain the advantages of **CHA** on the basis of the following three reasons:

First of all, "uptake" is a fundamental yet very important "function" in supramolecular chemistry. In addition, the uptake of *multiple*, large metal-complexes and organic compounds is still *unusual behavior*. To emphasize the unprecedented "improved uptake" abilities of micelle (**CHA**)_n, we added (i) "It should be noted that there has been no report on micelles as well as supramolecular containers (e.g., large hydrogen-bonding and coordination capsules)²⁵⁻²⁷ accommodating such multiple, bulky metal-complex and organic dyes" to page 5, (ii) "Whereas a huge number of fullerene-based 1:1 host-guest complexes have been reported so far, the facile preparation of host•(fullerene)_m complexes (m ≥ 2) remains challenging³⁰⁻³²" to page 6, and (iii) "The obtained host-guest complexes provide "multiple", large metal-complexes in the cavity, which is also uncommon for previously reported micelles and supramolecular containers" to page 8. In addition, we modified the following sentence: (iv) "To the best of our knowledge, ... there has been no report on emissive host-guest complexes including *multiple AuPz* compounds and its derivatives so far³⁶⁻⁴⁰." in page 7.

Second, we emphasize the unusual "uptake selectivity" of micelle (**CHA**)_n. Previously reported covalent, hydrogen-bonding, and coordination containers display the selective binding abilities toward *small and medium-sized* molecules, because of their rigid cavities. In contrast, it has been established that previous alkyl micelles as well as aromatic micelles possess flexible cavities so that the distinction between similar compounds, in terms of structure and properties, has been hardly achieved by the micellar containers so far. In this work, we *for the first time* revealed that micelle (**CHA**)_n provides uptake ability for perchlorinated Cu(II)-phthalocyanines but not for non-substituted/perfluorinated ones. To re-emphasized this result, we added "*Alkane-based*

micelles (e.g., $(\text{SDS})_n$ and its derivative) show poor uptake abilities and no selectivity toward **CuPc-Cl** under similar conditions²⁹.” to page 5.

Third, most importantly, we succeeded in the observation of strong red emission (>30% quantum yield) from $(\text{AuPz})_m$ in solution, through the efficient uptake of otherwise non-emissive **AuPz** compounds by micelle $(\text{CHA})_n$. We have already revealed and mentioned that “... no and low uptake of $(\text{AuPz})_m$ were observed with previous micelles $(\text{AA})_n$ and $(\text{SDS})_n$, respectively, under the same conditions” and “The emission intensity of the product is 3.8 times higher than that of $(\text{SDS})_n \cdot (\text{AuPz})_m$ ” in the main text (page 7). Trinuclear Au(I)-complexes are well-known, highly emissive compounds only in the solid state (exception: organogels), through weak intermolecular Au(I)•••Au(I) interactions. To the best of our knowledge, nobody could demonstrate the uptake-induced solution-state emission of the Au(I)-complexes by previous micelles as well as supramolecular containers so far.

These unusual host behavior, observed for the first time, is definitely derived from the design of a new micelle, composed of bent amphiphiles **CHA** with two cycloalkyl frameworks.

Finally, Reviewers #1 and #2 fully support the present work with “This is an interesting and well-performed study that would be of interest to the wide readership of *Nature Communications*... I believe the results of this manuscript will stimulate further investigation on this new class of micelles and I would recommend publication of the manuscript with the following minor corrections” and “This is a very interesting paper revealing novel types of micelles formed by small amphiphilic molecules with cycloalkane core segments. The micelles have unique complexation capabilities toward metal-containing molecules, allowing to preserve and further transform their photo-emission properties... I believe that the paper can be published in *Nature Communications*”.

On the basis of the three reasons (and the related improvement) and the full support from the two reviewers, we therefore believe that our present results provide enough novelty, in terms of molecular design and host function, for the publication in *Nature Communications*.

For example, the authors speculated that four C60 molecules were trapped together in the CHA amphiphile. Does the C60 aggregate show unusual redox behavior?

Thank you very much for this proposal. We are interested in the electrochemical properties of fullerene clusters and have already clarified the redox behavior of a C_{60} dimer in a coordination capsule (ref. 32). In addition, the Nitschke group has also revealed the redox behavior of $(\text{C}_{60})_m$ clusters ($m = 1-4$) in a coordination cage (ref. 30). However, for the CV studies of $(\text{CHA})_n \cdot (\text{C}_{60})_m$, we need to use organic solvents such as CH_3CN instead of water, due to the redox potential of the solvent. The host-guest structure of $(\text{CHA})_n \cdot (\text{C}_{60})_m$ remains only in water.

The CHA amphiphile can create a large hydrophobic space in water, where can accommodate several guest molecules. Did the authors try co-encapsulation of two different guests?

According to the reviewer’s suggestion for co-encapsulation, we additionally examined the uptake experiments of micelle $(\text{CHA})_n$ from mixtures of **ZnTPP/Rub**, **ZnTPP/CuPc-Cl**, **ZnTPP/C₆₀**, **C₆₀/SubPc**, and **C₆₀/CNT**, respectively. The UV-visible analysis of the products indicated the formation of a complex mixture of host-guest complexes (Supplementary Fig. 34). To further investigate the products’ properties, we need to improve the uptake efficiency and characterize the

detailed host-guest structures.

They will establish new triplet-triplet annihilation and donor-acceptor systems.

Thank you very much for these proposals again. Although we are also interested in the subjects, these are beyond the scope of the present work. The study of triplet-triplet annihilation is now in progress using “aromatic micelles”.

REVIEWERS' COMMENTS

Reviewer #1 (Remarks to the Author):

The authors have satisfactorily addressed the reviewers' comments with the exception of the report of multiplets as ranges. For example, the multiplets for overlapping proton signals c and d in 1,5-dicyclohexyl-2,3,4-trihydroxybenzene are not reported as a range. The authors have provided additional experiments regarding the uptake of mixture of guests. Although these experiments suggest the formation of a mixture of host-guest complexes rather than selective guest uptake, I believe the conclusions from the manuscript in its current form warrant its publication in Nature Communications.

Reviewer #3 (Remarks to the Author):

The authors kindly explained the novelty of this experiment and the paper was revised accordingly. Although the uptake experiments using two different guests, unfortunately, showed no clear evidence of co-encapsulation, this result may not matter at this stage. I would support this paper for publication.

For the comment of Reviewer #1:

the multiplets for overlapping proton signals c and d in 1,5-dicyclohexyl-2,3,4-trihydroxy benzene are not reported as a range.

We modified the suggested NMR data in the Methods session in the text.